# Vitamin C and Cardiovascular Disease: An Update

**DOI:** 10.3390/antiox9121227

**Published:** 2020-12-03

**Authors:** Marco B. Morelli, Jessica Gambardella, Vanessa Castellanos, Valentina Trimarco, Gaetano Santulli

**Affiliations:** 1Division of Cardiology, Department of Medicine, Institute of Aging Research, Wilf Family Cardiovascular Research Institute, Albert Einstein College of Medicine, Montefiore University Hospital, New York, NY 10461, USA; marco.morelli@einsteinmed.org (M.B.M.); jessica.gambardella@einsteinmed.org (J.G.); vanecastellanos35@gmail.com (V.C.); 2Department of Molecular Pharmacology, Fleischer Institute for Diabetes and Metabolism (FIDAM), Einstein-Sinai Diabetes Research Center (ES-DRC), Albert Einstein College of Medicine, New York, NY 10461, USA; 3Department of Advanced Biomedical Sciences, “Federico II” University, 80131 Naples, Italy; 4International Translational Research and Medical Education (ITME), 80100 Naples, Italy; 5Department of Neuroscience, “Federico II” University, 80131 Naples, Italy

**Keywords:** ascorbic acid, blood pressure, coronavirus, COVID–19, dietary supplements, drug formulations, endothelial dysfunction, GULO, heart failure, hypertension, infections, liposomes, oxidative stress, stroke, SVCT, vascular permeability, Vitamin C

## Abstract

The potential beneficial effects of the antioxidant properties of vitamin C have been investigated in a number of pathological conditions. In this review, we assess both clinical and preclinical studies evaluating the role of vitamin C in cardiac and vascular disorders, including coronary heart disease, heart failure, hypertension, and cerebrovascular diseases. Pitfalls and controversies in investigations on vitamin C and cardiovascular disorders are also discussed.

## 1. Physiology of Vitamin C

Vitamin C is a carbohydrate-like substance first isolated in 1928 by Albert Szent-Györgyi from cabbage, paprika, and the adrenal glands of some animals [1,2]; it was later found to be identical with the antiscorbutic factor that had been isolated from lemon juice by Sylvester Solomon Zilva in 1924 [3]. Originally called ‘hexuronic acid’ by Györgyi, the current denomination as vitamin C was proposed in 1933 by Walter Norman Haworth [4,5].

Among the vastly biological functions in which vitamin C is involved, as a potent antioxidant and radical scavenger, it protects cell constituents against oxidative stress, mediated by reactive oxygen species (ROS) and free radicals [6,7,8]. Moreover, vitamin C redox potential allows the maintenance of reduced and regeneration of other antioxidants, including glutathione and α–tocopherol (vitamin E) [9,10,11]. It is also required for the synthesis of several crucial biomolecules since it acts as a cofactor of the monooxygenase and dioxygenase enzymes [12]. Vitamin C-dependent enzymatic reactions are involved in the biosynthesis of collagen and cellular procollagen secretion, L-carnitine, norepinephrine, epinephrine, and for the regulation of the biosynthesis of other molecules [13,14]. Furthermore, vitamin C intake from food or supplements can increase the bioavailability of iron by improving the absorption of the non-heme iron [15,16,17,18].

## 2. Genetics and Diet

The crucial biological functions of vitamin C and the severity of the pathological consequences of its deficiency somehow explain the evolutive conservation of its biosynthesis within the great part of the mammalian class, as well as in many vertebrate animals [19]. However, this ancestral function has been lost during the evolution in humans, anthropoid primates, guinea pigs, and some species of monkeys and bats [20,21]. Albert Lehninger between the 1950s and 1960s was the first scientist to indicate that human cells are not able to perform the last reaction of vitamin C biosynthesis, i.e., the conversion of L-gulono-γ-lactone into ascorbic acid, which is catalyzed by the enzyme L-gulono-γ-lactone (GULO) oxidase [21,22,23]. It was only in 1991 that Nishikimi and colleagues demonstrated that the gene encoding for GULO is still present in humans but is not active due to the accumulation of a number of mutations that eventually turned it into a non-functional pseudogene [24,25]. Biochemists are continuing to investigate the exact reasons that led to the propagation of this inborn genetic flaw, namely the loss of a gene coding for an essential compound, during the evolution from a primate ancestor to humans. Although several mechanisms were proposed, the primary argument supporting the paradox that mammalian species with the loss-of-function mutations of the GULO gene were not under selective pressure is that all these species have diets that provide a high amount of vitamin C; therefore, the loss of their GULO gene did not cause any evolutive disadvantage [20,26,27,28]. This explanation is in agreement with the fact that wild anthropoid primates consume a considerably larger quantity of vitamin C per body weight compared to the human recommended daily amount for vitamin C (65 to 90 mg/day) [29]: dietary intake of vitamin C reported for gorillas (*Gorilla gorilla*) is 20–30 mg/kg/day, for spider monkeys (*Ateles geoffroyi*) is 106 mg/kg/day, and for howler monkeys (*Alouatta palliata*) is 88 mg/kg/day; similar considerations are valid for bats, with *Artibeus jamaicensis* consuming 258 mg/kg/day [20,28].

Based on the Third National Health and Nutrition Examination Survey (NHANES III, 1988–1994), the percentage of US males with vitamin C deficiency (less than 11.4 µmol/L in serum) was found to be greater compared to females, reaching a peak of 17% among 25- to 64-year-old subjects [30]. The more recent 2003–2004 NHANES revealed the one vitamin C status improved in the US population, and vitamin C deficiency was significantly lower than that observed during NHANES III; indeed, the overall prevalence of age-adjusted vitamin C deficiency was 7.1 ± 0.9% [31]. Yet, various studies have evidenced low vitamin C levels in both adults and children living in low-income countries [32,33,34,35]. In India, vitamin C deficiency has been relatively well characterized by large population-based studies of >5600 adults aged over 60, in which a marked deficiency was observed in 74% of adults living in north India and 46% of adults living in south India [32,36].

## 3. Pharmacokinetics and Bioavailability of Vitamin C

Vitamin C is a lactone (C_6_H_8_O_6_) and can exist both in a reduced and in an oxidized form: L-ascorbic acid (reduced form) and dehydro-L-ascorbic acid (oxidized form), as shown in Figure 1. Its hydroxyl groups are located at positions 2 and 3 ionize with pK values of 4.17 and 11.57. Although both forms are biologically active [37,38], in most body fluids, reduced vitamin C is mainly present as an ascorbate anion [39,40,41]. Due to its size and charge, ascorbate does not readily permeate the lipid bilayer [42,43,44]. The majority of intestinal uptake, tissue distribution, and renal reuptake is mediated by the Na^+^-dependent vitamin C transporters (SVCT1 and SCVT2), which co-transports Na^+^ and ascorbate across membranes: SVCT1 is widely distributed, while SVCT2 is the main isoform in the adrenal gland and in the eye [45,46,47,48,49,50,51,52,53,54,55].

Ingested vitamin C is mostly absorbed across the intestinal epithelium by membrane transporters in the apical brush border membrane, either as ascorbate via the SVCT transporters or as dehydroascorbic acid through facilitated diffusion via GLUT1 or GLUT3 transporters [56,57,58]; once inside the cell, dehydroascorbic acid is then converted to ascorbate or transported to the bloodstream by GLUT1 and GLUT2 in the basolateral membrane. Specifically, SVCT1 is mainly located on the apical membrane of enterocytes whereas SVCT2 is expressed at the basolateral surface [59,60,61,62,63,64]. SVCT1 is also present in the epithelium of the proximal renal tubuli [50], where it orchestrates the active reabsorption of ascorbate in the kidneys. A murine model lacking SVCT1 exhibited a significantly increased (18-fold) renal fractional excretion, though intestinal absorption was not reduced [65], indicating that the renal SVCT1-mediated reabsorption of vitamin C is crucial in determining its systemic homeostasis.

The non-linear pharmacokinetics and the unique compartmentalization of vitamin C at physiological levels [50,66,67,68,69] can be mostly attributed to the differences between organs in expression, concentration dependency, and substrate affinity of the SVCT and GLUT transporters [49,57,60,61,62,70,71,72]. In contrast to oral administration, parenteral administration of vitamin C bypasses the saturable transport mechanism that governs its intestinal absorption [73,74]. However, intravenous administration of vitamin C is an impractical mode of administration and carries risks of discomfort, infection, and phlebitis. Hence, alternative modes of effective vitamin C delivery have been proposed, with the ultimate goal of improving its bioavailability. In this regard, oral consumption of vitamin C encapsulated in liposomes may hold promise. Liposomes consist of single or multiple concentric membrane-like phospholipid bilayers encapsulating an aqueous compartment [75,76,77,78]. The size of these lipid vesicles can range from a few nanometers to several micrometers; liposomes specifically applied to medical use range between 50 and 450 nm [78,79,80,81]. Liposomes seem to be an ideal drug-carrier system, since their morphology is similar to that of cellular membranes and because of their ability to incorporate various substances, including hydrophilic molecules [82,83,84,85,86,87,88,89,90,91,92,93,94,95,96,97,98,99,100]. Accordingly, several studies have indicated that liposomes are very useful for delivering vitamin C [99,101,102,103,104,105,106,107,108,109,110,111,112,113,114,115]. For instance, Davis and coworkers evaluated plasma levels of oral, intravenous, and liposomal vitamin C, demonstrating that liposomal vitamin C has an enhanced bioavailability compared to non-liposomal vitamin C, while avoiding the risks associated with intravenous administration [116]. Another recent study has demonstrated that liposomal vitamin C increases the concentration of vitamin C in the blood by almost doubling the concentration obtainable via the non-liposomal form [117]. These results imply that liposomes might be an excellent carrier for vitamin C to achieve high blood levels.

## 4. Cardiovascular Effects of Vitamin C: Pre-Clinical Evidence

Severe vitamin C deficiency leads to a well-established disease, known as scurvy, a pathological condition characterized by bleeding, poor wound healing, hair and tooth loss, joint pain, and bone fragility [4]. Albeit scurvy is a fatal disease if not promptly treated, in several clinical studies, low levels of vitamin C (assessed by dietary intake or plasma analysis) have been associated with a number of conditions, including high blood pressure (BP), endothelial dysfunction, heart disease, atherosclerosis, and stroke [118,119,120,121]. Several pre-clinical studies investigated the role of vitamin C in cardiac and vascular protection and in the amelioration of pathological conditions. In 1999, Heller and collaborators demonstrated that ascorbic acid potentiates nitric oxide (NO) synthesis in cultured human endothelial cells, a mechanism that can preserve vessels from altered myogenic tone (vasoconstriction), atherosclerosis, and coagulation abnormalities [122]. Huang et al., in 2000, reported similar effects on the endothelium-derived NO synthase (eNOS) bioactivity, through the increase of the intracellular tetrahydrobiopterin content in porcine aortic endothelial cells [123]. These results were then confirmed by Baker et al., who reported a significant enhancement of eNOS enzymatic activity in human umbilical vein endothelial cells (HUVECs) treated with vitamin C, following the increase of the eNOS cofactor tetrahydrobiopterin [124]. 

Similar protective effects of vitamin C on vascular endothelial functions were reported in 2002 by d’Uscio et al., who observed that supplementation with ascorbate increases tetrahydrobiopterin and eNOS activity in the aortas of apoE-deficient and control mice [125]. Further evidence of the endothelial protection derived by vitamin C and tetrahydrobiopterin was provided later by Yan et al., who reported in a rat model that oral co-supplementation of vitamin C with tetrahydrobiopterin and L-arginine significantly increases vascular perfusion after hindlimb ischemia by augmenting eNOS activity and reducing oxidative stress and tissue necrosis [126]. 

In 2003, Ülker and collaborators reported that incubation of aortas from spontaneously hypertensive rats with different concentrations of vitamin C (10 to 100 μmol/L) improved endothelial function, reduced NAD(P)H oxidase activity, as well as superoxide production, and increased eNOS activity and NO generation to the levels observed in control rat aortas [127].

These observations were corroborated by Ladurner et al., who discovered that ascorbate rapidly activates eNOS in HUVEC and immortalized human endothelial cells, through a dose-dependent increased phosphorylation of eNOS-Ser^1177^ and concomitant decreased phosphorylation at eNOS-Thr^495^ [128], a phosphorylation pattern highly indicative of increased eNOS activity [129]. Glutathione in its reduced form (GSH) plays crucial roles in protecting cells from oxidative damage, as well as in maintaining cellular redox homeostasis [130,131]. GSH is essential for the physiological function of vitamin C because it is required in vivo for the reduction of dehydroascorbate in guinea pigs, unable to synthesize ascorbate [132]; moreover, GSH deficiency in adult mice stimulates ascorbate synthesis in the liver [132]. These findings strongly suggest the presence of a metabolic redundancy and overlap in the functions of these two important antioxidants [133].

While optimal levels of plasma vitamin C may be critical for maintaining glutathione in its reduced form, sparing it from depletion [134,135], the increment of the oxidized glutathione (GSSG) in the heart, with the consequent reduction of the GSH/GSSG ratio, could lead to ventricular arrhythmias via mechanisms that include mitochondrial depolarization in cardiomyocytes [136].

Overall, these observations indicate that vitamin C does not exert its antioxidative function only by direct interaction with ROS but also via maintenance of the redox regulation by increasing levels of other cellular radical scavengers.

## 5. Vitamin C and Cardiac Disease

Various clinical studies have investigated the association between supplementation of vitamin C and the risk of cardiovascular disease (CVD), heart failure (HF), coronary heart disease (CHD), and other major adverse cardiac events (Table 1 and Table 2), often yielding opposite results.

In 1993, Rimm et al. published a cohort study conducted on 39,910 US health professionals free of diagnosed CHD or other metabolic conditions at enrollment, and assessed by dietary questionnaires the intake of vitamin C, vitamin E, and carotene, showing that vitamin C (1162 mg/day) was not associated with a lower risk of CHD during a 4–year follow-up [137]. However, in 2003, a study conducted on 85,118 female nurses, based on a food-frequency questionnaire and using multivariate models to adjust for age, smoking, and other cardiovascular risk factors, indicated that the intake of vitamin C supplements was associated with a significantly lower risk of CHD during 16 years of follow-up [138].

These findings were confirmed in 2004 in a pooled analysis of nine prospective cohort studies, showing that supplementation of vitamin C (≥700 mg/day) was inversely associated with CHD risk during a mean 10–year follow-up [152]. 

Conversely, in the same year, a prospective (15–year follow-up) cohort study (the Iowa Women’s Health Study Cohort) found a positive association between mortality for CHD and vitamin C supplementation (≥300 mg/day), assessed via food-frequency questionnaires in diabetic postmenopausal women [153].

In 2008, a meta-analysis based on 14 studies, for a total of 374,488 individuals with a median follow-up of ~10 years, concluded that dietary vitamin C has an inverse association with CHD risk, whereas supplement intake of vitamin C has no significant association with CHD risk [154]. Another large prospective cohort study conducted for a median follow-up of 16.5 years found an inverse association between dietary vitamin C intake determined by a food frequency questionnaire and CHD mortality in Japanese women without a history of CVD or cancer [155]. In 2013, a meta-analysis evaluating 15 trials for a total of 188,209 subjects found that vitamin C (in a range from 120 to 1000 mg) in association with vitamin E and beta-carotene had no significant effect on CHD and major cardiovascular events [156].

Taken together, these analyses indicate that the association between vitamin C intake and CHD risk remains controversial, with variable association depending on the source of vitamin C. It is important to emphasize at this point that the relationship between vitamin C intake and the actual vitamin C levels in the blood is not linear, hence it is not appropriate to extrapolate vitamin C levels from food intake questionnaires, and values ascertained via blood analyses are necessary [32,72]. 

The better accuracy of the quantification of vitamin C levels from blood samples compared to food frequency questionnaires is further confirmed when analyzing the outcomes of the European Investigation into Cancer and Nutrition (EPIC)–Norfolk study, a prospective study investigating the relationship between vitamin C and incident HF in 20,299 healthy adults. After a median follow-up of ~12.8 years, ascorbate was inversely associated with incident cases of HF; specifically, every 20 μmol/L increase in plasma vitamin C concentration was associated with a 9% reduced risk of HF [121,139]. Interestingly, in this study, dietary intake was the primary source of vitamin C as assessed by the food frequency questionnaire; however, the investigators did not find a significant association between the reported consumption of fruit and vegetables and lower risk of HF. Therefore, food questionnaires are inadequate to establish the exact vitamin C levels of an individual, hence it is reasonable to consider blood samples as a more reliable source [32,72].

In 2007, the clinical trial Women Antioxidant Cardiovascular Study tested the effects of vitamin C, alongside beta carotene and vitamin E, on the combined outcome of myocardial infarction, coronary revascularization, or cardiovascular death among 8171 female health professionals aged 40 years or older, with a history of CVD or at least three CVD risk factors, followed up for ~9.4 years; the study found no significant effects of ascorbic acid (500 mg/day) on cardiovascular events [143]. Similarly, the Physicians Health Study II (PHS II), a large trial conducted on 14,641 male subjects, including 754 men (5.1%) with prevalent CVD at randomization, revealed that vitamin C (500 mg/day) for ~8 years of follow-up had no significant effect on cardiovascular mortality [147]. In 2017, a meta-analysis of eight randomized controlled trials with a total of 15,445 participants, found inconsistent or low-quality (downgraded) evidence for a correlation between vitamin C supplementation and major cardiovascular events or cardiovascular mortality [157].

In light of the contradictory outcomes herein discussed, it appears that better quality studies are necessary to examine the effects of vitamin C on cardiovascular endpoints, particularly in participants with different risks of CVD. Furthermore, study participants should be tested to exclude the contribution from singular or multiple micronutrient deficiencies, since vitamin C deficiency is commonly accompanied by other micronutrient deficiencies and associated with other confounding aspects, including poor-quality diet [32,158].

## 6. Vitamin C and Vascular Disease

We and others have shown that the endothelium plays a crucial role in the pathophysiology of vascular disorders [165,166,167,168,169,170,171,172,173,174,175,176,177]. Over the years, a significant number of clinical studies have investigated the effect of vitamin C on vascular disease, evidencing a promising association linking vitamin C intake, circulating levels of ascorbic acid, and lower risk of hypertension. Forman and coworkers reported data from three large prospective cohorts, namely the Nurses Health Study 1 (n = 88,540 women, median age 49 years), the Nurses Health Study 2 (n = 97,315 women, median age 36 years), and the Health Professionals Follow-up Study (n = 37,375 men, median age 52 years). 

The risk of developing hypertension did not substantially change for individuals who had reported an intake upper or equal to 1500 mg/day of vitamin C compared to those with an intake lower than 250 mg/day [178].

However, when the vitamin C plasma concentration was directly measured, three independent cross-sectional studies indicated a significant inverse correlation between the concentrations of ascorbic acid and BP values [119,120,179]. Moran et al. reported that vitamin C plasma concentrations in a group of 168 healthy US citizen were inversely related to systolic BP (SBP) [119]. Myint et al. published their observations based on the EPIC-Norfolk cross-sectional study, showing a strong relationship between plasma vitamin C concentrations and lower clinic BP; after adjusting for confounders, the likelihood of having high BP was 22% lower for subjects in the top quartiles of plasma vitamin C levels compared to the bottom quartiles [120]. 

Block et al. examined in a cross-sectional analysis the association of plasma ascorbic acid with BP in a group of 242 women (aged 18–21 years), finding that plasma vitamin C was inversely associated with both SBP and diastolic BP (DBP), highlighting how persons in the highest quartile of the plasma ascorbic acid distribution had 4.66 mmHg lower systolic BP than those in the lowest quartile of the distribution [179].

These results are consistent with a meta-analysis of 29 trials for a total of 1407 participants including normotensive and hypertensive subjects, showing that daily supplementation with a range from 60 to 4000 mg of vitamin C (median dose: 500 mg) in hypertensive participants reduced systolic BP by 3.84 mm Hg and DBP by 1.48 mm Hg [159].

In 2015, the Coronary Artery Risk Development in Young Adults (CARDIA), a prospective cohort study collecting data from 2884 young adults initially hypertension free, revealed that higher plasma ascorbic acid can favorably influence BP [149]. A very recent meta-analysis, published in 2020 by Ran et al., selected 11 cross-sectional studies and 7 case-control studies conducted from 1990 to 2017; the authors reported that serum vitamin C levels in the study participants were significantly lower in hypertensive subjects than in normotensive ones; moreover, the authors found a significant reverse relationship between serum vitamin C concentration and both SBP and DBP [162]. 

In the same year, an independent meta-analysis published by Guan et al. supported the results of Ran’s study: The authors selected 8 randomized controlled trials in which vitamin C, varying from 300 to 1000 mg/day, was supplemented for 4–24 weeks to 614 hypertensive patients; the authors observed a marked reduction in DBP in the subgroup of patients with an age ≥35 years having a vitamin C supplementation ≥500 mg/day for 6 weeks, while a significant reduction of both SBP and DBP was observed in the subgroup of age ≥60 years [163].

Overall, these findings highlight how vitamin C is inversely associated with both SBP and DBP, indicating that people with hypertension could exhibit low serum vitamin C compared to normotensive subjects. 

Bruno and colleagues demonstrated that acute vitamin C infusion (3 g *i.v.* in 5 min) significantly lowers BP in hypertensive patients but not in normotensive subjects; additionally, muscle sympathetic nerve activity was significantly reduced after vitamin C infusion in hypertensive patients but not in healthy subjects [180]. Corroborating these results, Heitzer et al. demonstrated that intra-arterial infusion of vitamin C (18 mg/min) had no significant effect on the forearm blood flow in response to acetylcholine (endothelium-dependent vasodilator) and sodium nitroprusside (endothelium-independent vasodilator) in 10 control subjects; however, in 10 chronic smokers, acetylcholine with the concomitant infusion of vitamin C markedly improved the impaired forearm blood flow responses [181]. Similarly, vitamin C supplementation (1000 mg for 4 weeks) significantly reduced LDL oxidative susceptibility in a group of smokers [182]. 

However, a subsequent clinical trial conducted on 34 male smokers did not support these observations; in fact, compared to placebo, supplementation of vitamin C (250 mg twice daily for 4 weeks) significantly increased plasma ascorbate concentrations but did not significantly change the levels of circulating oxidized LDL, nor of markers of endothelial activation like sICAM–1, sVCAM–1, and vWF-antigen [183].

In 2003, Salonen et al. published the results of the 6–year ASAP trial, in which slow-release vitamin C (250 mg) in addition to vitamin E (136 IU) were supplemented twice daily in hypercholesterolemic subjects [142]. Ultrasound assays proved that the vitamin C supplementation reduced the slope of the mean carotid artery intima-media thickness progression by 26%, with a significance that was reached in men (33% of reduction) but not in women (14% of reduction) [142]. These findings confirmed the observations previously obtained in the same ASAP trial at a shorter 3–year follow-up [184].

In 2014, two meta-analyses authored by Ashor and collaborators provided interesting findings concerning the link between vitamin C plasma levels and endothelial function, supporting the above-mentioned observations previously made by Heitzer [181] and Salonen [142]. The first meta-analysis, examining 44 randomized clinical trials comprising 1324 participants (964 males and 360 females) with a median age of ~51 years, revealed a significant positive effect of vitamin C supplementation (>500 mg/day) on endothelial function measured by flow-mediated dilation (19 studies) [161]. The action of vitamin C on endothelial function appeared to be dependent on health status, with stronger effects in atherosclerotic and diabetic patients. The second meta-analysis, performed on 22 randomized controlled trials, which included 1909 subjects (1088 males and 821 females) with an age that ranged from 22 to 63.5 (median of 29) years, reported that supplementation of vitamins C (120 to 4000 mg/day), E, A, and β–carotene improved arterial stiffness irrespective of the age group and duration of intervention [161]; notably, antioxidant vitamins were more effective in participants with low baseline plasma concentrations of vitamin C [161].

Recent trials have shown that Vitamin C ameliorates the oxidative imbalance and vascular remodeling induced by different stressors, including reperfusion injury following myocardial infarction [185], hyperoxya [186,187], shear stress [188], prolonged immobilization [189], glucose load [190], and mental stress [191,192]. Vitamin C also increases skeletal muscle blood flow and oxygen consumption via local vasodilation during exercise [193]. Vitamin C has also been shown to significantly attenuate endothelial barrier permeability [194,195,196], an aspect that has major implications in infectious disorders [197,198,199,200], which are also known to cause a systemic surge in oxidative stress [201,202,203]. 

Therefore, the antioxidant roles of Vitamin C and its protective effects on endothelial permeability could come into effect especially during infective processes [204,205,206,207]. Vitamin C has also been shown to improve the effects of other agents in a synergistic manner: for instance, when added to metformin, Vitamin C reduces cardiovascular diabetic complications [208], when added to glucagon-like peptide 1 (GLP–1), it further ameliorates endothelial function and reduces oxidative stress in Type 1 diabetic patients [209]. 

Thus, it is possible that the association of Vitamin C with other nutrients and micronutrients playing akin beneficial actions could be harnessed [210]. For instance, the supplementation of L-Arginine has been shown to play a favorable role in the regulation of immune responses [211,212]. Similarly, both Vitamin C and L-Arginine are known to improve endothelial function and reduce vascular permeability during infectious disorders [213,214,215,216]. Consequently, it is possible to speculate that their association could be synergistic in tackling infectious diseases; for instance, as COVID–19 is causing endotheliopathy [169], the association of oral L-arginine and liposomal vitamin C (e.g., Bioarginina^®^ C, 2 vials/day) could be efficacious for COVID–19 and other infectious disorders.

Several studies have assessed the association between Vitamin C and the potential protection from cerebrovascular disease, and the results have been subject of much controversy, similarly to what observed in the investigations assessing CHD. A prospective cohort study that followed for 20 years the residents of a rural Japanese community, having an age of 40 years or older and initially free of stroke diagnosis, has shown that higher serum Vitamin C concentration was associated with a reduced risk of incidence of both hemorrhagic and ischemic stroke [217]. Only a small part of these associations could be explained by lowering BP and adjusting for the physical activity level [217]. Similarly, in 20,649 adults followed for 10 years in the prospective cohort study EPIC-Norfolk, plasma ascorbate concentrations in the top quartile were linked to a 42% lower risk of stroke compared to the values in the lowest quartile [140]. 

Of note, in both the EPIC-Norfolk [140] and Japanese [217] populations, blood Vitamin C concentrations were positively correlated with vegetable and fruit intake. In 2011 Del Rio and colleagues investigated the relationship between dietary total antioxidant capacity and the risk of hemorrhagic and ischemic stroke in 41,620 subjects not previously diagnosed with myocardial infarction or stroke, recruited from the Italian segment of the European Prospective Investigation into Cancer and Nutrition (EPIC) study; the results from specific sub-analyses on the different stroke types indicate that Vitamin C is associated with a decreased risk of ischemic stroke [151]. These studies were consistent with a meta-analysis published in 2013, in which 16 prospective studies reported a 19% lower risk of stroke (pooled analysis of 2 studies) when comparing the highest *vs.* the lowest dietary Vitamin C intake, and a 38% lower risk of stroke (pooled analysis of 6 studies) when comparing the highest *vs.* the lowest circulating Vitamin C concentration [160]. 

On the other hand, Sesso and collaborators reported the results of a randomized double-blind placebo-controlled trial conducted in 14,641 healthy US male physicians, enrolled in the PHS II study at ≥50 years of age, showing that Vitamin C supplementation (500 mg/day) had no significant effect on the mortality from, or the incidence of, any type of stroke after a mean follow-up of ~8 years [156]. In agreement with these observations, Ye and collaborators published a meta-analysis study of 15 trials for a total of 188,209 participants, reporting that the supplementation of Vitamin C (in a range from 120 to 1000 mg) in association with Vitamin E and beta-carotene, had no significant effects on the risk of stroke compared to placebo [156]. 

## 7. Conclusions

Both preclinical and clinical data available in the literature show that Vitamin C plays a pivotal role in a number of processes involved in the pathogenesis of CVD. Nevertheless, major limitations have to be taken in consideration when interpreting the results of the studies investigating the effects of Vitamin C on human health; for instance, dietary assessment methods such as questionnaires or food diaries lack precision and accuracy and do not embrace other conditions that can affect Vitamin C levels and homeostasis [35,218,219,220,221,222]; therefore, these approaches are inadequate to ascertain the accurate Vitamin C status of an individual. Henceforth, it is reasonable to consider blood samples as more reliable indicators. Equally important, plasma Vitamin C level may considerably fluctuate following food or supplement ingestion. Consequently, the most reliable and practically available information on Vitamin C status can only be obtained from blood samples withdrawn in fasted conditions [32,72]. Ascorbate is rapidly oxidized *ex vivo* and the resulting oxidation products are quickly metabolized or degraded [38], hence, in order to obtain valid and consistent Vitamin C concentrations, a scrupulous sample handling is essential. Furthermore, the use of liposomal Vitamin C should be preferred to other forms since liposomes might be an excellent carrier for Vitamin C to achieve high blood levels.

Therefore, further studies are needed to clarify whether Vitamin C supplementation is actually effective in the prevention or cure of CVD. Specifically, prospective randomized clinical trials in large populations are needed to better define the best dose, the ideal administration route, the optimal targets, as well as the contribution of dietary supplementation of other antioxidant elements; in order to have standardized and reproducible data, it will be important to evaluate the Vitamin C status using blood samples obtained from fasted individuals.

## Figures and Tables

**Figure 1 antioxidants-09-01227-f001:**
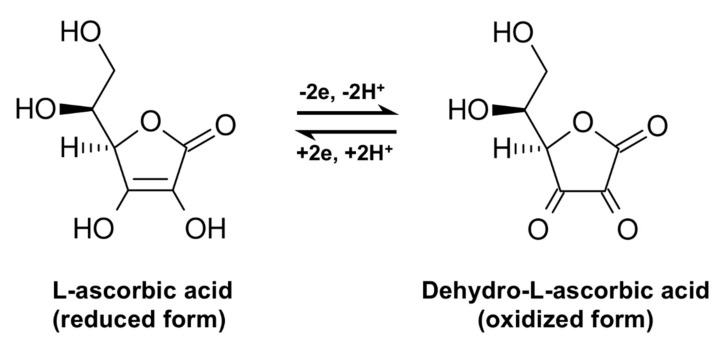
Reduced (left) and oxidized (right) forms of vitamin C.

**Table 1 antioxidants-09-01227-t001:** Major studies investigating the association between vitamin C and cardiovascular disease (CVD).

Clinical Study	Study Population	Vitamin C Concentration Range	Fasting Blood Sample (*Assay Used*)	Adjustments	Outcome	Ref.
**EPIC-Norfolk**	19,496 healthy men and women from Norfolk (US)	25.9–79.2 μmol/L	Non-fasting venous blood (*fluorometric assay*)	Age, systolic BP, cholesterol, smoking, diabetes, and supplement use	A mean follow-up of 4 years showed that 20 μmol/L rising in plasma Vitamin C concentration was associated with ~20% reduced risk of all-cause mortality	[121,139]
**EPIC-Norfolk**	20,299 healthy men and women from Norfolk (US)	41–66 μmol/L	Non-fasting venous blood (*fluorometric assay*)	Age, sex, smoking, BMI, physical activity, systolic BP, diabetes, cholesterol, MI, social class, alcohol, and supplement used	A mean follow-up of 12.8 years showed that every 20 μmol/L increase in plasma Vitamin C was associated with a 9% relative reduction in the risk of HF.	[140]
**HPFS—Health Professionals Follow-up Study**	39,910 healthy US male, health professionals	95–116 mg/dL	Not reported	Age, season, smoking, energy intake, alcohol, hypertension, history of MI, profession, BMI, and physical activity	After a mean follow-up of 8 years, Vitamin C intake was not associated with a lower risk of CHD and stroke	[141]
**NHS I—Nurses’ Health Study 1**	85,118 healthy US female nurses	70–704 mg/dL	Not reported	Age, energy intake, supplements use, alcohol, smoking status, and diabetes	Data from 16 years of follow-up indicate that Vitamin C supplement is associated with a significantly lower risk of CHD	[138]
**ASAP—Antioxidant Supplementation in Atherosclerosis Prevention**	520 men and post-menopausal women, with hyper-cholesterolemia	Not specified	Not specified (*HPLC assay*)	Sex, baseline HDL cholesterol	A 6–year follow-up indicated that Vitamin C supplementation reduced the slope of the mean carotid artery intima-media thickness progression	[142]
**Women’s Antioxidant Cardiovascular Study**	8171 female health professionals with CVD, or 3 or more CVD risk factors	Not specified	Not reported	Compliance-adjusted analyses	After a follow-up of 9.4 years, no overall effects of Vitamin C (500 mg/day) on all major cardiovascular events and stroke	[143]
**NHANES II**	8417 US men and women	17–79.5 μmol/L	Non-fasting venous blood (*colorimetric assay*)	Age, sex, race, education, physical activity, dietary energy and fat intake, cholesterol, BMI, alcohol, smoking, diabetes, hypertension, CHD, stroke, cancer, use of aspirin, Vitamin E intake	Decreased risk of all-cause mortality	[144]
**NHANES III**	6624 US men and women	17–85.2 μmol/L	Non-fasting venous blood (*colorimetric assay*)	Age, sex, race, education, physical activity, smoking, alcohol, cholesterol, BMI, diabetes, hypertension, use of aspirin, Vitamin E intake	Top Vitamin C group had a 27% lower CVD risk compared to bottom group	[145]
**SU.VI.MAX study**	13,017 French adults	Not specified	Fasting venous blood (*continuous flow segmented by air bubbles*)	Bonferroni adjustment for multiple mean comparisons	After a follow-up of 7.5 years, no effect of Vitamin C supplementation (120 mg/d) on CVD; lower all-cause mortality in men	[146]
**PHS II—Physicians Health Study II**	14,641 healthy US male physicians	Not specified	Not reported	Age, PHS cohort, randomized β–carotene, Vitamin E or Vitamin C, and multivitamin assignments	A mean follow-up of ~8 years showed that Vitamin C supplementation (500 mg/day) had no significant effect on the incidence of CVD, stroke, or mortality for cardiovascular events	[147]
**Heart Protection Study**	20,536 British men and women with CHD, occlusive arterial disease, or diabetes	Not specified	Non-fasting venous blood	Not specified	After a follow-up of 5 years, Vitamin C supplementation (250 mg/d) had no significant effects on CVD	[148]
**CARDIA**	2884 healthy US men and women	24–70 μmol/L	Fasting venous blood (*HPLC assay*)	Age, sex, race, center, education, smoking, alcohol, physical activity, *a priori* diet quality score and food consumption, BMI, waist circumference, history of diabetes, systolic BP	Higher plasma Vitamin C, due to better diet quality, is inversely related to high BP.	[149]
**Iowa Women’s Health Study**	1923 post-menopausal diabetic women from U.S. (*55 to 69 years)*	82–≥679 mg/day	Not reported	Age, total energy intake, history of hypertension, BMI, waist-hip ratio, smoking status and amount smoked, alcohol consumption, physical activity score, hormone replacement therapy, main type and duration of diabetes, medications	After a follow-up of 11 years, an association between increased mortality for CVD or CHD and Vitamin C ≥ 300 mg/day was reported. Vitamin C intake was unrelated to CVD and stroke mortality in nondiabetic women.	[150]
**Italian section of the EPIC Study**	41,620 Italians previously diagnosed with stroke or myocardial infarction (*44 to 61 years*)	83–201 mg/day	Not reported	Age, sex, hypertension, smoking, alcohol, education, energy intake, waist circumference, obesity, and physical activity	After 7.9 years of follow-up, Vitamin C was associated with a lower risk of ischemic stroke	[151]

**Table 2 antioxidants-09-01227-t002:** Main meta-analyses investigating the correlation between vitamin C intake or its plasma levels and cardiovascular disease (CVD).

Meta-Analysis Topic	Number of Studies	Total Participants (n)	Trial Duration	Vitamin C Supplement Dosage	Outcome	Ref.
Coronary Heart Disease	14 cohorts	374,488	mean duration: 10 y.	Not reported	Dietary Vitamin C had an inverse association with CHD risk, but supplement intake of Vitamin C had no significant association with CHD risk.	[154]
Hypertension	29 trials	1407	mean duration: 8 wks.	60 to 4000 mg/day (median 500 mg/day)	Daily supplementation of Vitamin C in hypertensive participants reduced systolic BP by 3.84 mm Hg and DBP by 1.48 mm Hg.	[159]
Coronary Heart Disease	15 trials	188,209	from 1.4 y. to 12 y.	120 to 1000 mg/day	Vitamin C in association with Vitamin E and beta-carotene had no effect on CHD or other major cardiovascular events. Vitamin C in association with Vitamin E and beta-carotene had no effect on stroke risk compared to placebo.	[156]
Stroke	16 cohorts	217,454 participants for studies on Vitamin C intake 29,648 participants for studies on circulating Vitamin C	from 6.1 y. to 30 y.	45 to 1167 mg/day	19% lower risk of stroke (pooled analysis of 2 studies) comparing the highest to the lowest dietary Vitamin C intakes. 38% lower risk of stroke (pooled analysis of 6 studies) with the highest versus lowest circulating Vitamin C concentrations.	[160]
Endothelial function	44 trials	1324	from 1 d. to 8 wks.	500 to 2000 mg/day	A significant positive effect of Vitamin C supplementation on endothelial function measured by flow-mediated dilation (19 trials), evaluating pulse-wave (5 trials) and forearm blood flow (20 trials).	[161]
Arterial stiffness	22 trials	1909	from 1 d. to 7 y. (mean duration: 56 d.)	120 to 4000 mg/day (mean 2000 mg/day)	Association of Vitamin C with Vitamin E and β–carotene improved arterial stiffness irrespective of the age group	[161]
Cardiovascular events	8 trials	15,445	mean duration: 8 y.	–	Inconsistent or low-quality (downgraded) evidence for the correlation between Vitamin C supplementation and major cardiovascular events or cardiovascular mortality	[157]
Hypertension	11 cross-sectional studies and 7 case-control studies	22,200	–	–	Serum Vitamin C level was significantly lower in hypertensive compared to normotensive subjects. Inverse correlation between serum Vitamin C concentration and SBP or DBP.	[162]
Hypertension	8 trials	614	from 4 wks. to 24 wks.	300 to 1000 mg/day	Significantly reduced SBP and DBP in the subgroups of age ≥35 and ≥60 years with Vitamin C supplementation ≥500 mg/day for 6 weeks or more.	[163]
Cardiovascular mortality	15 trials	320,548	from 4 to 22y.	50 to 450 mg/day	Higher Vitamin C intake is associated with a lower risk of cardiovascular mortality	[164]

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
