# Peer review of "Vitamin C and Cardiovascular Disease: An Update"

_antioxidants, 2020, doi:10.3390/antiox9121227_

Round 1
Reviewer 1 Report
The review paper concerning the potential role of vitamin C in human cardiovascular diseases (CVD) in its broadest terms provides a very basic review of the functions of the vitamin but does not relate these to the data presented from the survey studies that are summarized (but in no way synthesized). There is a short paragraph concerning liposomes but no clinical data is provided, nor are any laboratory studies cited using vitamin C-containing liposomes and vascular tissue, as just one potential example. There are very few references to research from 2010 onwards. Section 9 is unrelated to the topic. The authors need to answer the question: what new information does this review bring to the field of either vitamin C research or CVD research?
Author Response
WE THANK THIS REVIEWER FOR HER/HIS USEFUL REMARKS, WHICH CONTRIBUTED TO IMPROVE THE QUALITY OF OUR MANUSCRIPT.
WE ADDED MORE REFERENCES TO RECENT STUDIES (THE REVISED VERSION OF THE MANUSCRIPT INCLUDES MORE THAN 100 REFERENCES OF PAPERS PUBLISHED AFTER 2010;
SECTION 9 HAS BEEN REMOVED.
OUR REVIEW REPRESENTS AN UPDATED OVERVIEW OF THE FUNCTIONAL ROLE OF VITAMIN C IN CARDIOVASCULAR DISEASE, INCLUDING BOTH PRECLINICAL AND CLINICAL STUDIES AND INNOVATIVE DELIVERY APPROACHES LIKE LIPOSOMES.
Reviewer 2 Report
Review report
The review by Morelli et al. addresses an interesting and relevant issue: namely, an assessment of clinical and preclinical studies evaluating the role of vitamin C in cardiac and vascular disorders, including coronary heart disease, heart failure, hypertension and the modulation of vascular permeability in systemic disorders.
There are several issues that should be addressed prior to publication:
3 bioavailability of vitamin C (line 78 ff)
In the paragraph, i.v. administration is explained to guarantee the highest plasma concentration of vitamin C compared to oral administration. Please describe in more details the pharmacokinetics of vitamin C including the individual transporter systems accountable for uptake, reuptake and distribution of ascorbic acid from intestine, in the kidneys and in tissue - and explain how their activity dependent on the plasma concentrations of vitamin C. The plasma vitamin C concentrations are highly important to take into consideration when interpreting the effects of oral vitamin C supplementation in clinical intervention studies.
Furthermore, it is be relevant to mention the transporters responsible for the distribution of vitamin C from the plasma to the tissue.
Please also add the mechanisms of liposomal-encapsulated vitamin C uptake compared to non-liposomal vitamin C – and explain why the formulation augments the bioavailability of vitamin C?
4 Cardiovascular effects of Vitamin C: Pre-clinical evidence (line 112 ff)
The paragraph (line 154 to 162) about l-carnitine refers to clinical – and not pre-clinical evidence, as written in the paragraph heading.
The presentation of the clinical studies is in generally very referring, with limited reflections on why the effects of vitamin C are highly variable between the studies. Do the clinical studies suffer from selection bias with e.g. an overweight of well-nourished persons? How often - and how - is the vitamin C intake estimated? Is plasma vitamin C samples from fasted blood samples? Which methods are used to assess plasma vitamin C (e.g. HPLC, colorimetric methods)? Does the patients in the placebo group suffer from vitamin C deficiency or do they have optimal vitamin C status - in which way could that have an impact on the effect of vitamin C? Are the patients in the placebo group allowed concurrent supplementation?
In general, it would be very vulnerable if the authors could structure and add the above listed information for the individual studies in in tables and reflect on potential weaknesses of the studies.
Author Response
WE THANK THIS REVIEWER FOR HER/HIS COMMENTS AND SUGGESTIONS.
PHARMACOKINETICS OF VITAMIN C IS NOW DETAILED IN A DEDICATED SECTION, AS REQUESTED, AS WELL AS THE TRANSPORTERS RESPONSIBLE FOR THE UPTAKE AND DISTRIBUTION OF VITAMIN C. A PARAGRAPH ON LIPOSOMES EXPLAINING THEIR ROLE IN INCREASING THE BIOAVAILABILITY OF VITAMIN C HAS BEEN ADDED AS WELL.
WE THANK THIS REVIEWER FOR THE PERTINENT COMMENT. IN THE REVISED VERSION OF THE MANUSCRIPT, WE NOW DISCUSS THE POTENTIAL MECHANISMS UNDERLYING THE DIFFERENCES IN THE OUTCOMES OF THE VARIOUS STUDIES. WE ALSO PROVIDE A REVISED TABLE 1 THAT INCLUDES IMPORTANT INFORMATION ON THE METHODS USED TO ASSESS PLASMA VITAMIN C, SPECIFYING WHETHER THE BLOOD SAMPLES WERE WITHDRAWN IN FASTING CONDITIONS.
Reviewer 3 Report
A well-written review article covering both pre-clinical and clinical data.
The authors discuss potential roles in the treatment of infectious diseases. Vitamin C is also one of the hot topics in septic shock management, and several studies have been published recently. It'll be more helpful to add these.
Author Response
THANKS.
WE HAVE ADDED SOME RECENT STUDIES ON VASCULAR PERMEABILITY IN SEPSIS.
Round 2
Reviewer 1 Report
The authors have provided a comprehensive updated review of the association of vitamin C status ( as determined by plasma levels rather than diet) and a number of clinically relevant cardiovascular outcomes. The authors may further strengthen their review by adding a column to Table 1 that reviews the actual findings in each study. Also, the tables could have subgroups showing those studies that measured plasma vitamin C. The authors may want to comment on what future studies should include in addition to plasma C levels.
Author Response
WE THANK THIS REVIEWER FOR HER/HIS COMMENTS AND SUGGESTIONS.
WE HAVE MODIFIED OUR MANUSCRIPT ACCORDINGLY.
Reviewer 2 Report
The authors have improved the manuscript, however there are still some parts that have to be clarified.
line 70 - 121: please also include the SVCT2 transporter somewhere in the text.
line 92 - 95: The authors refer to papers where vitamin C (> 5g) have been given intravenously and write that elimination adheres to 1st order kinetics with a constant half-life of about 2 hours. Please clarify in which plasma concentration span vitamin C elimination adheres to this kinetics and clarify the relevance of the kinetics in the context of the clinical studies refereed to in table 1 and 2.
line 105 ff: Are liposomes relevant in context of the clinical studies refereed to in table 1 and 2? If not, please shorten the paragraph.
Line 118: Please clarify "therapeutic plasma levels of vitamin C". I would be preferable if the authors could clearly distinguish between pharmacological effects of vitamin C (above physiological levels), and the effects of vitamin C to avoid deficiency.
Conclusion is quite fragmented. Please rephrase it.
Author Response
The authors have improved the manuscript, however there are still some parts that have to be clarified.
WE THANK THIS REVIEWER FOR HER/HIS COMMENTS AND SUGGETSIONS.
line 70 - 121: please also include the SVCT2 transporter somewhere in the text.
SVCT2 ARE NOW DISCUSSED, THANKS.
line 105 ff: Are liposomes relevant in context of the clinical studies refereed to in table 1 and 2? If not, please shorten the paragraph.
WE HAVE SHORTENED THE PARAGRAPH ON LIPOSOMES (IT HAD BEEN EXPANDED UPON REQUEST OF ANOTHER REVIEWER).
Line 118: Please clarify "therapeutic plasma levels of vitamin C". I would be preferable if the authors could clearly distinguish between pharmacological effects of vitamin C (above physiological levels), and the effects of vitamin C to avoid deficiency.
THAT SENTENCE WAS INCLUDED IN THE PARAGRAPH DISCUSSIN LIPOSOMES, AND HAS NOW BEEN REMOVED
The authors refer to papers where vitamin C (> 5g) have been given intravenously and write that elimination adheres to 1st order kinetics with a constant half-life of about 2 hours. Please clarify in which plasma concentration span vitamin C elimination adheres to this kinetics and clarify the relevance of the kinetics in the context of the clinical studies refereed to in table 1 and 2.
THE SENTENCE ON 1st ORDER KINETICS HAS BEEN REMOVED AS WELL.
Conclusion is quite fragmented. Please rephrase it.
WE HAVE MODIFIED THE CONCLUSION SECTION, AS REQUESTED.